**Data Availability Statement:** All relevant data are within the paper and its Supporting Information files.

# CTDSP1 inhibitor rabeprazole regulates DNA-PKcs dependent topoisomerase I degradation and irinotecan drug resistance in colorectal cancer

Hiroya Matsuoka[1], Koji Ando[1]*, Emma J. Swayze[2], Elizabeth C. Unan[2], Joseph Mathew[2], Quingjiang Hu[1], Yasuo Tsuda[1], Yuichiro Nakashima[1], Hiroshi Saeki[3], Eiji Oki[1], Ajit K. Bharti[2], Masaki Mori[1]

1 Department of Surgery and Science, Graduate School of Medical Sciences, Kyushu University, Fukuoka City, Fukuoka, Japan, 2 Division of Hematology and Medical Oncology, Department of Medicine, Boston University School of Medicine, Boston, Massachusetts, United States of America, 3 Department of General Surgical Science, Graduate School of Medicine, Gunma University, Maebashi, Gunma, Japan

* k-ando@surg2.med.kyushu-u.ac.jp

## Abstract

Irinotecan specifically targets topoisomerase I (topoI), and is used to treat various solid tumors, but only 13–32% of patients respond to the therapy. Now, it is understood that the rapid rate of topoI degradation in response to irinotecan causes irinotecan resistance. We have published that the deregulated DNA-PKcs kinase cascade ensures rapid degradation of topoI and is at the core of the drug resistance mechanism of topoI inhibitors, including irinotecan. We also identified CTD small phosphatase 1 (CTDSP1) (a nuclear phosphatase) as a primary upstream regulator of DNA-PKcs in response to topoI inhibitors. Previous reports showed that rabeprazole, a proton pump inhibitor (PPI) inhibits CTDSP1 activity. The purpose of this study was to confirm the effects of rabeprazole on CTDSP1 activity and its impact on irinotecan-based therapy in colon cancer. Using differentially expressing CTDSP1 cells, we demonstrated that CTDSP1 contributes to the irinotecan sensitivity by preventing topoI degradation. Retrospective analysis of patients receiving irinotecan with or without rabeprazole has shown the effects of CTDSP1 on irinotecan response. These results indicate that CTDSP1 promotes sensitivity to irinotecan and rabeprazole prevents this effect, resulting in drug resistance. To ensure the best chance at effective treatment, rabeprazole may not be a suitable PPI for cancer patients treated with irinotecan.

## Introduction

Topoisomerase I (topoI) was identified as a specific target for camptothecin (CPT) and its analogues like irinotecan and topotecan [1]. Irinotecan is frequently used to treat colon, gastric, ovarian, pancreatic, breast, and small cell lung cancer. TopoI reduces DNA supercoiling by cutting and re-ligating DNA, and a controlled rotation between the cutting and re-ligation

**Funding:** The authors received no specific funding for this work.

**Competing interests:** The authors have declared that no competing interests exist.

cycles reduces the supercoiling. However, in the presence of irinotecan, the re-ligation cycle is inhibited, and collision of progressing replication fork with nicked DNA leads to a DNA double strand break (DNA-DSB) and cell death [2]. CPTs are used extensively to treat various solid tumors, however, only 13–32% of patients respond, and the mechanisms of resistance are not well understood [3]. Classical mechanisms of irinotecan resistance potentially involve an ATP-binding cassette (ABC) transporter, ABCG2, which reduces intracellular drug accumulation [4–6]. Inhibition of the ABCG2 drug efflux pump using sorafenib sensitizes both non-resistant cells and resistant cells to irinotecan [7, 8]. Another proposed mechanism for irinotecan resistance is topoI gene mutation [9]. However, DNA sequencing studies have failed to find topoI gene mutations in cancer patients [10, 11]. Thus, none of the proposed drug resistance mechanisms have been validated. In response to irinotecan, topoI is ubiquitinated and degraded by ubiquitin proteasomal pathway (UPP) and rapid topoI degradation is shown to be associated with irinotecan resistance [12]. However, the precise mechanism was not understood and molecular determinants were not defined. Recently we have published the mechanism of topoI degradation by UPP, and have shown that the DNA-PKcs kinase cascade determines the rate of topoI degradation. Importantly, the deregulated kinase cascade keeps the DNA-PKcs constantly active, leading to higher basal levels of phosphorylated topoI-S10 and rapid degradation of topoI that leads to irinotecan resistance. To identify the upstream regulator of DNA-PKcs, we performed a siRNA library screen of 56 reported nuclear phosphatases, and subsequently phosphatase-silenced cells were treated with irinotecan to determine the CPT-induced rate of topoI degradation. Carboxy-terminal domain RNA polymerase II polypeptide small phosphatase 1 (CTDSP1) was identified as one of the strongest upstream regulators of DNA-PKcs-dependent proteasomal degradation of topoI.

The DNA-dependent protein kinase (DNA-PK) is a serine/threonine protein kinase composed of a large catalytic subunit (DNA-PKcs) and Ku 70/80 heterodimer [13]. It is now established that the Ku 70/80 heterodimer binds broken DNA double strands and recruits DNA-PKcs in vitro [14]. Once recruited, DNA-PK stabilizes the broken strand, targets Artemis for end processing, and finally the XLF/XRCC4/Ligase IV complex carries out the ligation. This classic NHEJ is the major eukaryotic pathway for DNA double strand break repair [15]. Although DNA-PKcs is considered a component of the DNA-double strand repair (DDR) pathway, recent findings indicate a variety of other important roles of this kinase in genome maintenance and tumor pathogenesis [16]. However, our understanding of the role of DNA-PKcs in tumor pathogenesis is not complete, and regulation of DNA-PKcs kinase activity is only partly understood. Autophosphorylation at multiple S/T sites is the first mechanism of DNA-PKcs kinase activation and inactivation, particularly in response to DNA-DSB. The second regulatory component is dephosphorylation of DNA-PKcs by phosphatases [13].

DNA-dependent RNA polymerase II consists of 12 polypeptides. The largest polypeptide Rpb1 contains heptad repeats, (Tyr-Ser-Pro-Thr-Ser-Pro-Ser) in the c-terminus domain (CTD), this unique feature of RNAPII distinguishes it from other polymerases [17]. The 52-heptad tandem repeats of CTD determine the regulation of transcription. Post-translational modifications of the 7 amino acid tandem repeats, particularly Ser2 and Ser5, are considered critical in RNAP II—dependent mRNA transcription. The pattern of CTD phosphorylation during the transcription cycle is highly dynamic and requires the activity of dedicated phosphatases and kinases [17]. Several kinases have been identified to phosphorylate CTD, most notably CDK7, CDK8, and CDK9. FCP1, Ssu72, and small CTD phosphatases (SCP) dephosphorylate the CTD during the different phases of MRNA transcription. CTDSP1 is part of the SCP (Small CTD Phosphatases) family, whose loss of function results in tumor development and proliferation [18]. This nuclear phosphatase associates with RNAP II and de-phosphorylates the serine 5 of the heptad repeat [19, 20]. The phosphorylation state of RNAP II affects its

transcription activity; therefore CTDSP1 plays a role in gene expression regulation [20]. Recent studies have shown that the activity of CTDSP1 is suppressed by the proton pump inhibitor, rabeprazole [21].

In this study, we demonstrated that CTDSP1 promotes sensitivity to irinotecan and rabeprazole prevents this effect, resulting in drug resistance.

# Materials and methods

## Cell culture and drug treatment

The human colon cancer cell lines HCT116, HT29, DLD1, and LoVo were obtained from ATCC. HCT116 and DLD1 cells were grown and maintained in RPMI containing 10% FBS and pen-strep. HT29 cells were grown and maintained in McCoy's medium containing 10% FBS and pen-strep. LoVo cells were grown and maintained in Ham's F12-K containing 10% FBS and pen-strep. All cells were grown at 37 ˚C and 5% $CO_2$ in a humidified cell culture incubator. Topo I inhibitor treatment was performed using various concentrations of either irinotecan (Sigma-Aldrich) or SN-38 (Tocris). SN-38 is the irinotecan active metabolite. Cells were also treated with various concentrations of Rabeprazole (Santa Cruz Biotechnology).

## CTDSP1 siRNA transfection

CTDSP1-specific siRNA (Silencer Pre-designed siRNA: 5′–CCUCGUGGUUUGACAACAU– 3′) and negative control were purchased from Dharmacon. Transfection of HCT116 cells ($0.6 \times 10^6$ cells/well in 6-well plate) with siRNA oligonucleotides was performed using Lipofectamine RNAiMAX, following the manufacturer's instructions (Invitrogen).

## CTDSP1 overexpression

For reverse transcription quantitative RT-PCR, the following mRNA sequences were used: CTDSP1, forward primer, 5′–CACCATGGACAGCTCGGCCGTCATTACTC–3′, and reverse primer, 5′–CTAGCTCCCTGGCCGTGGCTGCCTG–3′. The cDNA was synthesized from a total RNA of HT29 cells using Super Script 3 First-Strand Synthesis (Invitrogen). Quantitative PCR amplification was performed using a C1000 Touch Thermal Cycler. The PCR product and the pENTR/D-TOPO vector were mixed and incubated for 5 min at room temperature. The reaction was used to transform competent *E. coli* cells (Mach1, Invitrogen). The recombinant bacteria were screened using an LB agar medium containing kanamycin. The recombinant plasmid pENTR-CTDSP1 was extracted from positive colonies using the QIAprep Spin Miniprep Kit (Qiagen) and proper orientation of the cloned fragment was verified by PCR and DNA sequencing. The recombinant plasmid was transfected into HT29 cells using 4D-Nucleofecor (Lonza).

## Immunoblotting

Cells cultured in 6-well plates were scraped into an ice-cold RIPA buffer. Samples were clarified by centrifugation at 15000 rpm for 15 min at 4˚C. We used the iBind Western System (Invitrogen) and performed the imaging using the Amersham Imager 600 instrument (GE Healthcare, Little Chalfont, UK). Primary antibodies included antibodies against β-actin (Cell Signaling Technology), CTDSP1 (Abcam), Topoisomerase I (BD Pharmingen), and DNAPKcs-pS2056 (Abcam). Antigen/antibody complexes were visualized by enhanced chemiluminescence (ECL detection system).

## Immunofluorescence

Control and CTDSP1 knockdown cells were grown on sterile coverslips in 100mm plates and after washing with PBS, cells were fixed with 3.7% formaldehyde. After 25 minutes of fixation, coverslips were washed with 0.2% Triton-X-100 and then blocked with 3% BSA for 1 hour. After blocking, each coverslip was incubated with a monoclonal anti-phosphorylated DNAPK antibody, followed by Alexa- fluor 488-conjugated goat anti-rabbit IgG. Nuclei were stained with DAPI and imaging was performed on the Leica SP5 fluorescence microscope.

## Integrating EGFP following the *hTOPI* gene in HCT-116 cells

Summary sequences of plasmids used in this study can be found in S1 Table. Roughly 1000 bases 5' and 3' of the genomic sequence flanking the last *hTOP1* exon in HCT-116 cells were sequenced to identify and account for any cell-type specific polymorphisms. To do so, multiple PCR products spanning the genomic sequence were amplified using Phusion DNA Polymer-ase (New England Biolabs). The resulting PCR amplicons were cloned using the Zero Blunt TOPO PCR Cloning Kit (Invitrogen), transformed into E.coli XL-1 blue competent cells, and ~20–25 colonies were grown overnight at 37˚C in TB media prior to miniprep and sanger sequencing. A single guide-RNA (sgRNA) targeting the *hTOP1* stop codon was designed so that the SpCas9 binding site would be destroyed following gene conversion. To generate the sgRNA plasmid, oligonucleotides corresponding to the spacer sequence of the target site were annealed and ligated into BsmBI cut BPK1520. The homologous recombination donor plas-mid designed to create the topoI-EGFP fusion protein was generated by Gibson assembly into the NheI and HindIII sites of pUC19. Regions of the genomic sequence 5' and 3' of the *hTOP1* stop codon were assembled with an EGFP-P2A-*Pac* (for puromycin resistance) fusion cassette to generate the final donor plasmid (with 5' and 3' homology regions of 934 and 824 bases, respectively). Transfections into HCT-116 cells were performed by lipofectamin (lifetechnol-ogy) method with: 1) 2µg of a wild-type *Streptococcus pyogenes* Cas9 expression plasmid (MSP469); 2) 1µg of the sgRNA expression plasmid (MMW134) and 3) 2µg of the homologous recombination donor plasmid (MMW274). After 7 days of the transfection, cells were selected with 4µg/ml puromycin and after 14 days of selection, the cells were sorted by MoFlo Legacy (Beckman Coulter). The sorted cells were maintained in 2µg/ml puromycin contained media. Precise incorporation of the genome editing cassette in single cell cloned HCT-116 *hTOP1*-EGFP cells was confirmed by PCR amplifying the genomic locus and sequencing (S1 Data).

## Cell viability assay

Cells were seeded in 96-well plates at 700–2000 per well in 180 µl of medium. Next, a 100 µM SN-38 stock solution was used to prepare dilutions in 20 µl of medium before treatment. After incubating for 72 h, cells were incubated CellTiter-Glo 2.0 reagent into each well. Cell viability was then measured by luminescence detection using a Synergy H1 microplate reader (BioTek).

## Clonogenic assay

All cells were treated with various concentrations of SN-38 (0, 2.5, 5.0, or 7.5 nM) for 24 h, and then 50 cells per well were seeded in 6-well plates. After 14 days, when colonies were apparent, cells were fixed in 6% glutaraldehyde and 0.5% crystal violet for 15 min, then washed with tap water. The colonies per well were counted.

## Patients

Retrospective data from 61 patients who underwent 2nd-line chemotherapy including irinotecan between January 2003 and December 2015 at the Department of Surgery and Science, Kyushu University, were retrieved. Of the 61 patients, 5 patients who discontinued chemotherapy, or whose data was not complete, were excluded from the study. Finally, 56 patients were eligible for analysis.

## Statistical analysis

Each blots were quantified using Image J software (NIH, Bethesda, MD, USA). The statistical analyses were performed by using the JMP 14 statistical software package (SAS institute, Cary, NC, USA). The Student t-test, the chi-square test, Fisher's exact test and ANOVA one-way test were used where appropriate.

## Study approval

All experiments were conducted with approval by the Ethics Committee of the Kyushu University (Approval Number: No. 30–215)

## Results

### CTDSP1 expression correlated with topoI degradation in colorectal cancer cell line

We have previously reported that topoI is phosphorylated by DNA-PKcs at serine-10, and the resulting phosphoprotein (topoI-pS10) is efficiently ubiquitinated and targeted for rapid proteasomal degradation [3]. Protein phosphatases have been shown to interact with DNA-PKcs and could serve as upstream regulators of DNA-PKcs. To identify potential upstream regulators of DNA-PKcs, we screened a siRNA library comprised of all known human nuclear phosphatases. Our data demonstrated that silencing of CTDSP1 significantly altered the rate of topoI degradation. In the present study, the possible role of CTDSP1 as a regulator of topoI stability and irinotecan sensitivity was tested first by assessing the relationship between CTDSP1 expression and rate of topoI degradation. Cell lysates from four CRC cell lines, HCT116, DLD1, LoVo, and HT29 were subjected to immunoblot analysis with anti-CTDSP1. The results indicate a higher level of CTDSP1 in HCT116 cells and the lowest level in HT29 (Fig 1A). HCT116 and HT29 cells were treated with SN-38 for 90 and 180 min and cell lysates were subjected to determine the rate of topoI degradation. The lower level of topoI protein at 180 min clearly indicates a higher rate of topoI degradation in HT29 cells compared to HCT116 cells (Fig 1B). We then asked the obvious question: does this SN-38-induced rate of topoI degradation have an impact on drug sensitivity? The clonogenic survival assay clearly demonstrated that HT29, with low CTDSP1 and rapid topoI degradation, is more resistant to HCT116 cells (Fig 1C).

### Silencing of CTDSP1 enhances topoI degradation and irinotecan resistance

We asked if reducing CTDSP1 expression would change the rate of topoI degradation and impart cellular resistance to SN-38 in HCT116 cells. CTDSP1 expression was reduced significantly by siRNA (Fig 2A). Control and siCTDSP1- HCT116 cells were treated with SN38 and harvested at 90 and 180 min. The cell lysates were analyzed by immunoblotting with anti-topoI and anti-β-actin (for protein loading). The quantitative analysis of the immunoblot demonstrated a statistically significant reduction in topoI level in siCTDSP1-HCT116 cells. No

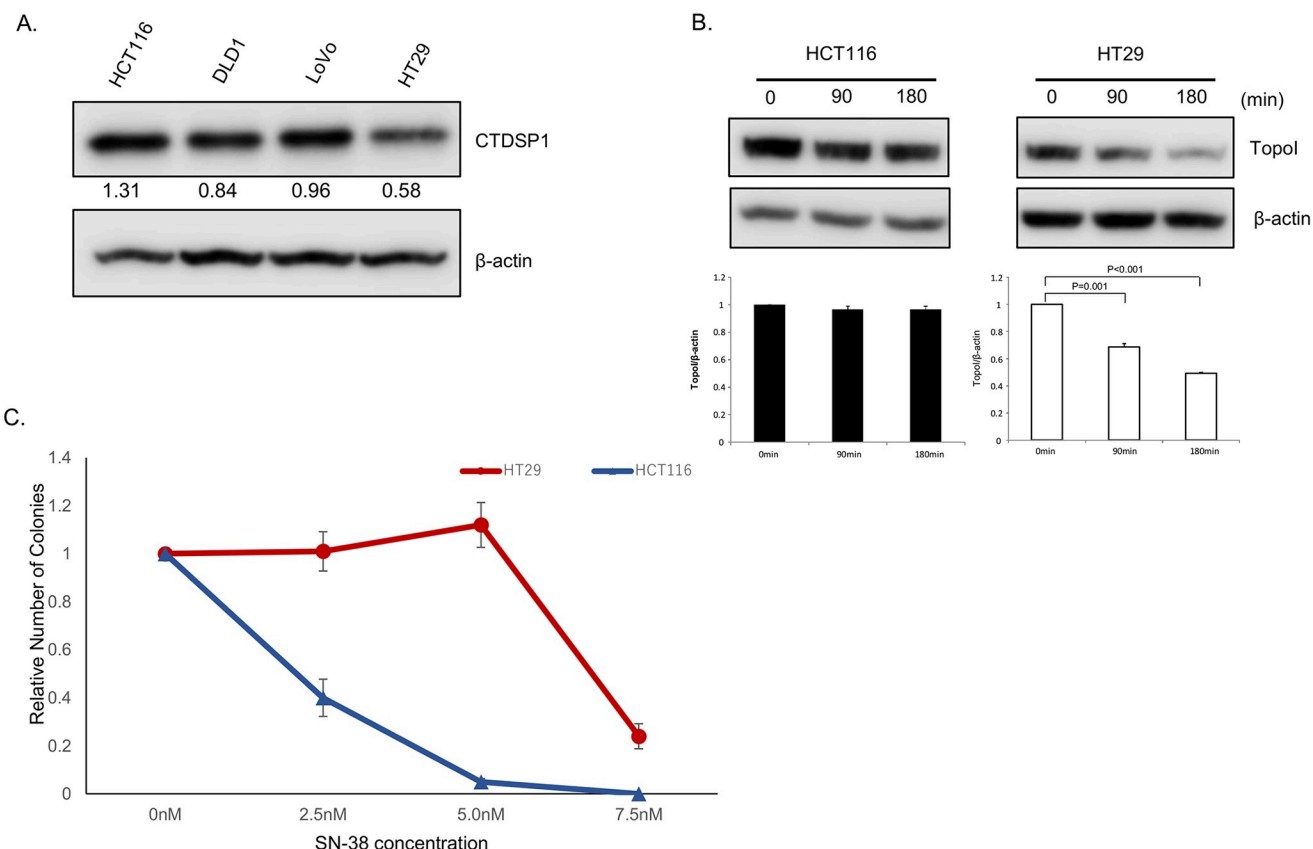

**Fig 1. CTDSP1 regulates irinotecan sensitivity in colon cancer cell lines. A**, Cell lysates from four colon cancer lines, HCT15, DLD1, LoVo and HT29, were immunoblotted with anti-CTDSP1 and anti-β-actin. **B**, HCT116 and HT29 cells were treated with 2.5 μM SN-38 and harvested after 90 and 180 min. Cell lysates were immunoblotted with anti-topoI and anti-β-actin. **C**, HCT116 and HT29 cells were treated with various concentrations of SN-38 and clonogenic assays were performed to determine the relative number of colonies.

appreciable reduction in topoI protein level was observed in control-HCT116 cells (Fig 2B). To further understand the CTDSP1-mediated rate of topoI proteasomal degradation in response to SN38, we silenced CTDSP1 in genomically edited HCT116 cells expressing topoI-GFP protein. Cells were treated with SN38 and relative florescence of topoI-GFP was visualized by confocal microscope. Untreated control and siCTDSP1 cells demonstrated similar topoI-GFP florescence, however, when cells were treated with SN-38, siCTDSP1 cells had lower topoI-GFP florescence, indicating a higher rate of degradation (Fig 2C). Comparative clonogenic assays and cell survival assays demonstrated that siCTDSP1 was significantly more resistant to SN38 (Fig 2D–2F).

### The higher expression of CTDSP1 in HT29 cells inhibits topoI degradation and restores SN-38 sensitivity

The basal level of CTDSP1 protein expression in HT29 is low compared to other CRC cells and SN38-induced proteasomal degradation in these cells is high. To better understand the role of CTDSP1 in SN-38-induced proteasomal degradation of topoI, we overexpressed CTDSP1 in HT29 cells. The stably transfected HT29 cells showed significantly higher CTDSP1 compared to vector-only cells (Fig 3A). Vector-only (control) and CTDSP1 overexpressing HT29 cells were treated with SN-38 and the cells were harvested at 90 and 180 min. The cell

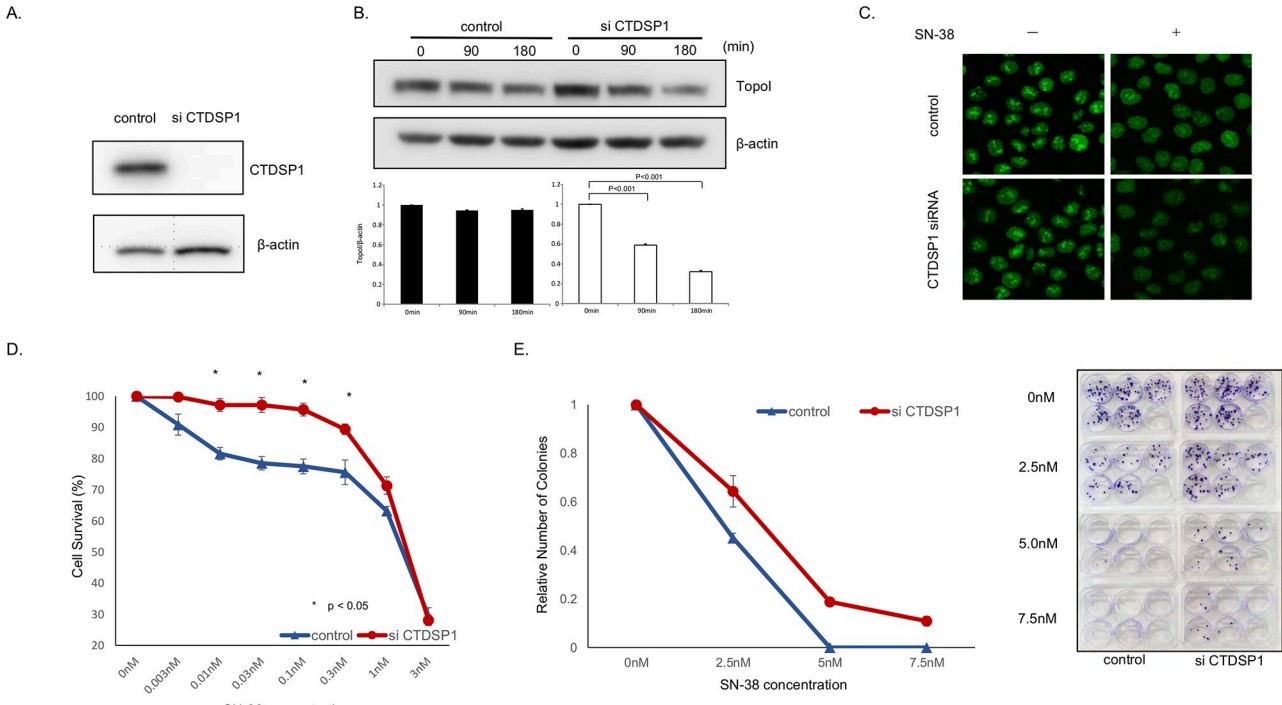

**Fig 2. Silencing of CTDSP1 enhances topoI degradation and irinotecan resistance. A**, Cells transfected with CTDSP1 or control siRNA were lyzed and the cells' lysates were immunoblotted with anti-CTDSP1 and anti-β-actin antibodies. **B**, HCT116-siRNA CTDSP1 or control siRNA, treated with 2.5 μM SN-38 were harvested after 90 and 180 min. Cell lysates were immunoblotted with anti-topoI and anti-β-actin antibodies. Cells' lysates were immunoblotted with anti-topoI and anti-β-actin antibodies. **C**, EGFP was integrated with topoI in HCT116 cells using CRISPR/Cas9 system and CTDSP1 was knocked down in this cell line by siRNA. Cells were treated with 2.5uM SN-38 for 60 min and the topoI-EGFP signal was imaged by Leica SP5 confocal microscope. **D**, HCT116-siRNA-CTDSP1 or control siRNA were plated in a 96-well plate and treated with various concentrations of SN-38 for 72 h. Cell viability was determined by detecting the luminescence. **E**, HCT116-siRNA-CTDSP1 or control siRNA cells were plated in a 6-well plate and treated with various concentrations of SN-38 for 24 h. Then, 50 cells per well were plated in a 6-well plate. After 14 days, when colonies were apparent (right panel), colonies were counted and the relative number of colonies was determined (left panel).

lysates were immunoblotted with anti-topoI and anti-β-Actin. The topoI proteins in vector alone cells were significantly reduced both at 90 and 180 min after SN-38 treatment. In contrast, very little or no topoI degradation was observed in cells overexpressing CTDSP1 (Fig 3B). A comparative estimation of cell survival assays demonstrated a significantly increased SN-38 sensitivity in cells overexpressing CTDSP1. The drug sensitivity difference was most pronounced at 10 nm SN-38 concentration (Fig 3C).

## CTDSP1 activates DNA-PKcs and enhances SN-38 induced proteasomal degradation of topoI and drug resistance

Recently we have published that DNA-PKcs dependent phosphorylation of topoI at Serine 10 is critical for SN-38- induced proteasomal degradation of topoI. We have also demonstrated that CTDSP1 is one of the upstream effector nuclear phosphatases that regulates DNA-PKcs activation and topoI proteasomal degradation. To better understand the role of CTDSP1 in this pathway, we first asked if deferential expression of CTDSP1 in HCT116 and HT29 affects DNA-PKcs kinase activity. Phosphorylation of DNA-PKcs-pS2056 is indicative of activated DNA-PKcs kinase. Cell lysates from HCT116 and HT29 cells were immunoblotted with anti-DNA-PKcs-pS2056 and the data clearly demonstrated a higher phosphorylation status of S2056 only in HT29 cells. Importantly HCT116 cells demonstrated minimal DNA-PKcs kinase

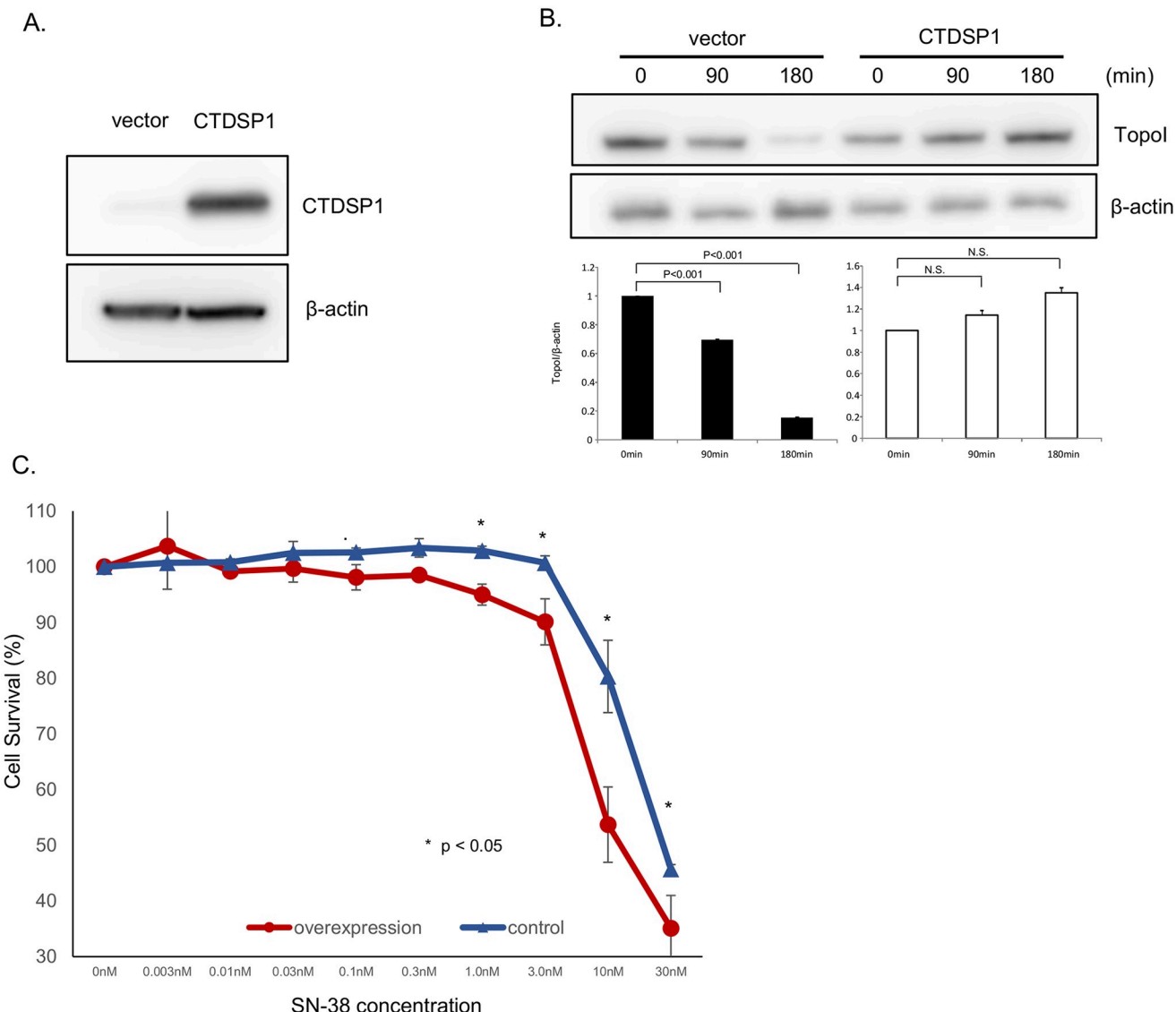

**Fig 3. CTDSP1 restores irinotecan sensitivity in HCT116 cells. A**, CTDSP1-overexpressing and control cells were lyzed and the cell lysates were immunoblotted with anti-CTDSP1 and anti-β-actin antibodies. **B**, CTDSP1-overexpressing and control cells were treated with 2.5 µM SN-38 and harvested after 90 or 180 min. Cells lysates were immunoblotted with anti-topo I and anti-β-actin antibodies. **C**, CTDSP1-overexpressing and control cells were plated in a 96-well plate and treated with various concentrations of SN-38 for 72 h. Cell viability was determined by luminescence detection.

activity. The GAPDH immunoblotting indicated that a similar amount of protein was analyzed (Fig 4A). To better understand the CTDSP1 dependent activation of DNA-PKcs, DNA-PKcs-S2056 phosphorylation status was determined in CTDSP1-silence HCT116 cells. The cells were also treated with SN38 to determine the effect of DNA-PKcs activation due to SN-38-induced DNA-DSB. CTDSP1 silencing resulted in remarkable activation of DNA-PKcs and there was not much difference in control and treated cells (Fig 4B). Control and siCTDSP1-HCT116 cells were also immunoflorescentely stained with anti-phosphospecific DNA-PKcs, and results indicated a higher basal level of phospho-DNA-PKcs compared to control cells (Fig 4C).

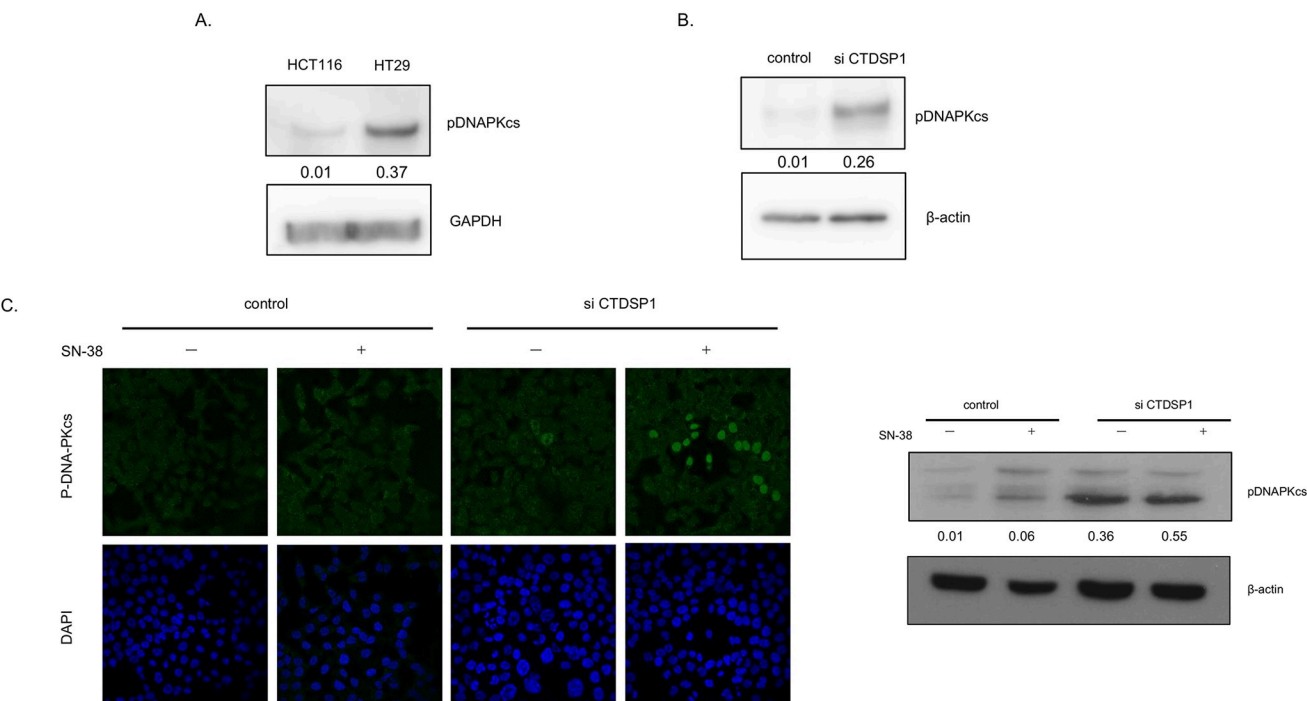

**Fig 4. CTDSP1 activates DNA-PKcs and enhances DNA-PKcs dependent topoI degradation in response to irinotecan. A**, HCT116 and HT29 cell lysates were immunoblotted with anti-DNAPKcs-pS2056 and anti-GAPDH antibodies. **B**, CTDSP1 was silenced by CTDSP1 siRNA in HCT116 cells, and control as well as CTDSP1-silenced cell lysates were subjected to immunoblot with anti-CTDSP1 and anti-β-actin. **C**, Control and CTDSP1 knockdown cells were treated with SN38 and analyzed by immunofluorescence staining with anti-DNA-PKcs–pS2056). The cells were analyzed by Leica SP5 confocal microscope.

## Rabeprazole inhibits CTDSP1 and enhances topoI degradation and irinotecan resistance

Rabeprazole binds to the hydrophobic pocket of CTDSP1 and inhibits its phosphatase activity. This hydrophobic pocket is adjacent to the active site in CTDSP1 and rabeprazole most likely acts as a direct competitor of the natural substrate, i.e., the CTD phosphorylated peptide [21]. We treated HCT116 cells with rabeprazole and then with SN-38. Cells were harvested at 90, 180, and 270 min post-SN-38 treatment, and cell lysates were analyzed to determine the rate of topoI degradation. The results indicate that the topoI degradation in response to SN-38 increased when cells were pretreated with rabeprazole (Fig 5A). The quantitative analysis of the immunoblot demonstrated significantly lower topoI protein at 90 min that continued til the 270 min time point. The rate of topoI degradation was enhanced when cells were either pretreated with 10 or 20μm rabeprazole. However, the degradation rate was more pronounced in cells pretreated with 20μm rabeprazole. In contrast, no appreciable topoI protein degradation was observed in HCT116 cells that were not pretreated with rabeprazole (Fig 5A). To further confirm these findings, genomically edited HCT116 cells with topoI-GFP protein were first treated with 5 and 10 μm of rabeprazole, and after 48 h the cells were treated with SN-38. The relative fluorescence intensity of GFP was observed by confocal microscope to determine SN-38-induced rate of topoI degradation. The data indicates that the rate of topoI degradation increased when cells were pretreated with rabeprazole. GFP fluorescence intensity was lower when cells were pretreated with 10μm compared to 5μm rabeprazole. Importantly cells not

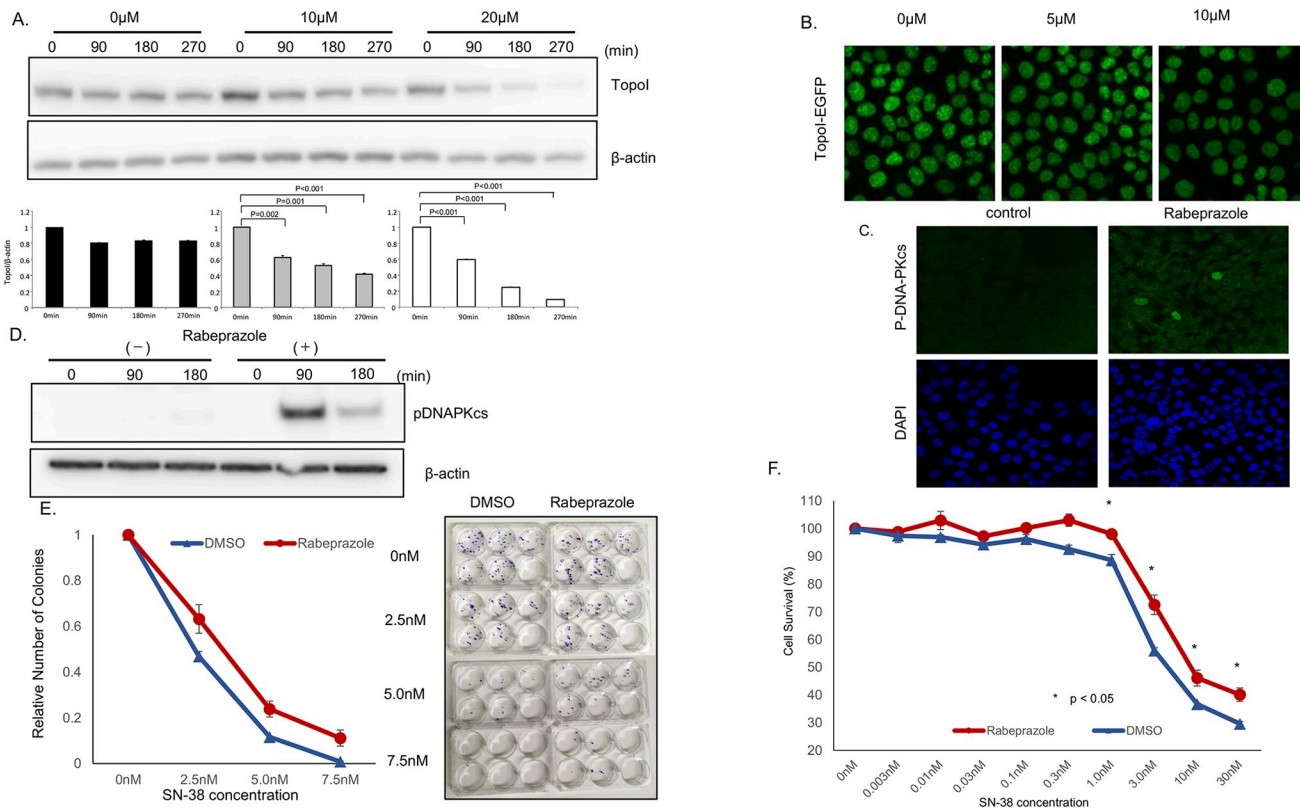

**Fig 5. Rabeprazole promotes topo I degradation and irinotecan resistance. A**, HCT116 cells were plated in a 6-well plate and treated with various concentrations of rabeprazole (0, 10, 20 μM) for 72 h, and then with 2.5 μM SN-38 and harvested after 90 or 180 min. Cell lysates were immunoblotted with anti-topoI and anti-β-actin. **B**, Genomically edited HCT116 cells with TopoI-EGFP fusion proteins were treated with 5 and 10 μM of Rabeprazole for 48 hours and topoI-GFP protein level was analyzed by confocal microscope. **C**. HCT116 cells were plated in a 6-well plate, treated with 40 μM rabeprazole or DMSO for 72 h, and then with 2.5 μM SN-38, and harvested after 90 or 180 min. Cell lysates were immunoblotted with anti-pDNA-PKcs and anti-β-actin. **D**, HCT 116 cells were treated with rabeprazole, control and treated cells were analyzed by immunofluorescence analysis with anti-phospho-DNA-PKcs-pS2056 and confocal microscopy. **E**. HCT116 cells were plated in a 6-well plate and treated with rabeprazole or DMSO for 72 h. Then, 50 cells were plated in each well of a 6-well plate and treated with various concentrations of SN-38 for 24 hours. Cell colonies were counted after 14 days. **F**. HCT116 cells were plated in a 6-well plate and treated with 40 μM rabeprazole or DMSO for 72 h. Then, cells were plated in a 96-well plate and treated with various concentrations of SN-38 for 72 h. Cell viability was determined by luminescence detection.

pretreated with rabeprazole topoI-GFP protein level remained high in the response to SN-38 (Fig 5B).

To further understand this pathway and the role of rabeprazole in SN-38-induced topoI degradation, immunofluorescence analysis of control and rabeprazole-treated HCT116 cells were immunostained with anti-DNA-PKcs-pS2056. Cells treated with rabeprazole demonstrated higher immunofluorescence levels, indicating kinase activation in response to rabeprazole (Fig 5C). Rabeprazole-treated cells were also analyzed by immunoblotting analysis with anti-DNA-PKcs-pS2056. The data clearly demonstrated the activation of DNA-PKcs at 90 min post SN38 treatment. The activation was transient and phosphorylated S2016 of DNA-PKcs was reduced at 180 min. Immunoblot analysis with anti-β-actin showed equal protein loading. To determine the effect of rabeprazole-induced proteasomal degradation of topoI and irinotecan resistance, HCT116 cells were treated with rabeprazole, and after 48 hours, cells were treated with DMSO (control) or various concentrations of SN-38 and clonogenic and cell survival assays were performed. The results demonstrated that the cells treated with rabeprazole

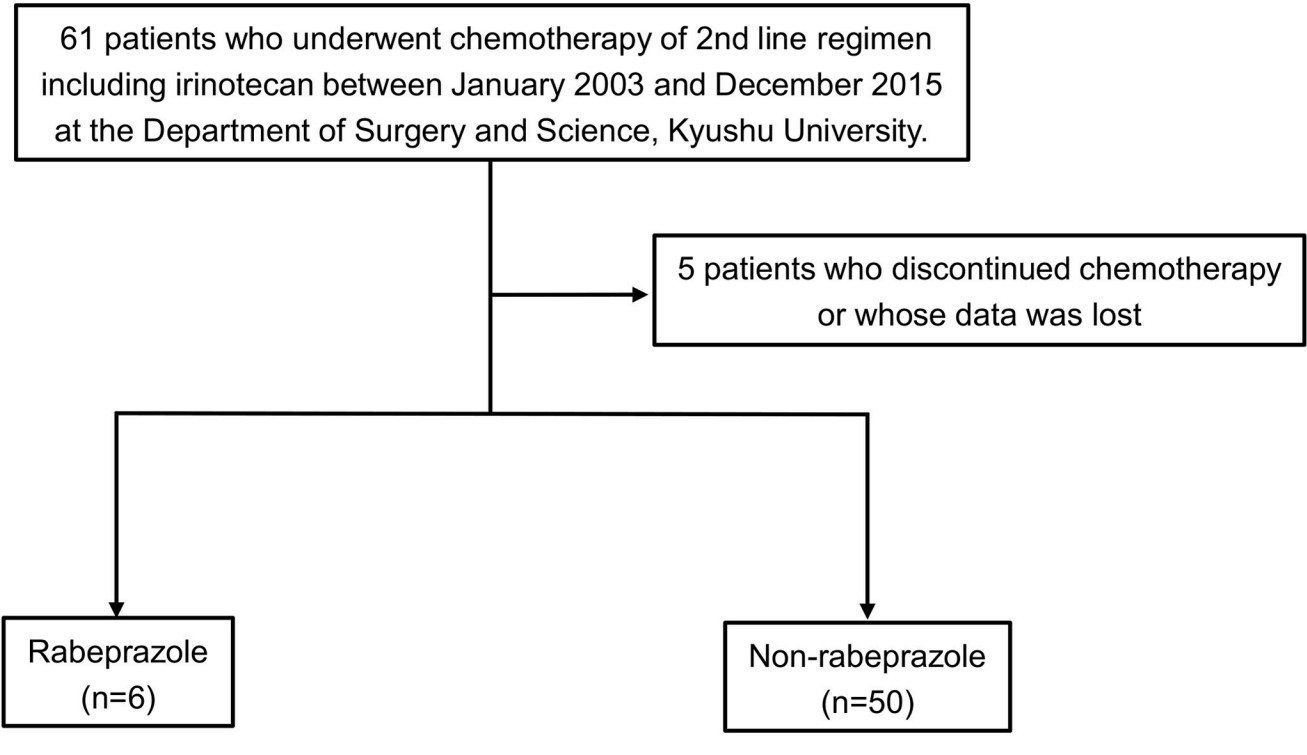

**Fig 6. Flow chart depicting the process of patient selection.**

had a higher number of colonies, and a higher percentage of cells was viable in rabeprazole-treated cells. The difference is statistically significant (Fig 5E and 5F).

## Treatment with rabeprazole may reduce the effect of irinotecan in colorectal cancer patients

We next examined clinical data to investigate rabeprazole-induced irinotecan resistance in patients with colon cancer. Of 56 eligible patients who underwent chemotherapy with irinotecan, 50 (89.3%) had not received rabeprazole during or before chemotherapy, while 6 (10.7%) had been treated with rabeprazole (Fig 6). We compared the overall response in the two groups. Partial response (PR) was observed in only one patient (16.7%) of the rabeprazole group, while 14 (28%) in the non-rabeprazole group had PR (Table 1). Similarly, the stable disease (SD) percent was higher and progressive disease was lower in patients not receiving

**Table 1. The overall response in the rabeprazole and non-rabeprazole group.**

|  | Best overall response | |
|---|---|---|
|  | **Rabeprazole** | **Non-rabeprazole** |
| CR | 0(0%) | 1(2.0%) |
| PR | 1(16.7%) | 14(28.0%) |
| SD | 2(33.3%) | 22(44.0%) |
| PD | 3(50%) | 13(26.0%) |

RECIST: Response Evaluation Criteria Inn Solid Tumors; CR, complete response; PR, partial response; SD, stable disease; PD, progressive disease.

rabeprazole. The results suggest that rabeprazole had an effect on the response to irinotecan in clinical settings.

## Discussion

One of the most remarkable phenomena observed in the cellular response to irinotecan is degradation of topoI. It was first observed in leucocytes of patients treated with 9-nitro-camptothecin in phase 1 clinical trials. The leucocytes isolated after 24, 48, and 72 hours demonstrated decreasing topoI protein levels [22]. Later, these findings were reproduced in cancer cells [23]. More importantly, Desai et al, in a series of publications, demonstrated that the irinotecan-induced degradation of topoI is mediated by a ubiquitin proteasomal pathway, cells degrade topoI differentially, and the cells that degrade topoI rapidly are resistant to irinotecan [24]. More recently we have demonstrated that: i) topoI associates with DNA-PK, and DNA-PKcs phosphorylates topoI at Serine 10; ii) Phosphorylated topoI is ubiquitinated by BRCA1; iii) cells with higher basal levels of topoI-pS10 degrade topoI rapidly and are resistant to irinotecan; iv) the higher basal level of topoI-pS10 is maintained by phosphatase dependent activation of DNA-PKcs v) nuclear phosphatase siRNA library screen identified PTEN and CTDSP1 enhances irinotecan-induced topoI degradation and vi) silencing of PTEN enhanced DNA-PKcs activity and irinotecan resistance [3]. Phosphorylation-dependent activation and inactivation of DNA-PKcs is well documented. Autophosphorylation of several S/T moieties localized in ABCD and PQR clusters is at the core of regulation of DNA-PKcs kinase activity in response to DNA-DSB and phosphorylation of serine 2506 indicates the activation of DNA-PKcs [13]. The second proposed mechanism of DNA-PKcs regulation depends on the dephosphorylation by phosphatases. In one of the early reports, protein phosphatase 1 or protein phosphatase 2A were shown to reactivate autophosphorylated inactive DNA-PKcs. Furthermore, the addition of a phosphatase inhibitor reverses this process [25]. PP2A was also shown to play a role in NHEJ by directly dephosphorylating DNA-PKcs [26]. Another phosphatase, PP6, was shown to regulate the kinase activity of DNA-PKcs in response to DNA-DSB. Both catalytic units (PP6C) and regulatory subunits (PPR1) of PP6 interact with DNA-PKcs, and silencing of PP6C induced IR sensitivity and delayed release from the G2/M checkpoint [27, 28].

Our siRNA library screen of nuclear phosphatase identified PTEN and CTDSP1 as two phosphatases that significantly enhance CPT-induced topoI degradation. Importantly, we have demonstrated that PTEN regulates DNA-PKcs kinase activity in this pathway and PTEN deletion ensures DNA-PKcs dependent higher topoI-pS10, rapid topoI degradation and irinotecan resistance [3]. We asked if CTDSP1 also dephosphorylates DNA-PKcs, and is the upstream regulator of CPT-induced topoI degradation. CTDSP1 is a SCP1 family of phosphatases that dephosphorylates RNAP II and plays a regulatory role in mRNA transcription. Our novel finding indicates that CTDSP1 dephosphorylates DNA-PKcs, changes its kinase activity, and regulates irinotecan-induced topoI degradation. Our library screen demonstrated that silencing of CTDSP1 enhances irinotecan-induced topoI degradation in HCT15 cells. HCT15 cells degrade topoI rapidly and are resistant to irinotecan. To determine the activation of this pathway in cells that do not degrade topoI, we used WT and genomically edited HCT116 cells. We impaired CTDSP1 phosphatase activity either by silencing by siRNA or using a specific inhibitor. The results clearly demonstrated that either CTDSP1 inhibition or silencing caused the activation of DNA-PKcs, indicated by phosphorylation of DNA-PKcs-S-2056. Importantly, the knocking down of CTDSP1 resulted in higher basal phosphorylation of DNA-PKcs-S-2056. Similar results were observed when cells were treated with rabeprazole. The activated state changed in response to irinotecan when the DNA-PKcs-S2056 protein level was analyzed

by immunoblotting and immunofluorescence. One of the important physiological functions attributed to DNA-PKcs in irinotecan-induced pathways is the degradation of topoI. HCT116 cells do not degrade topoI, however, a rapid topoI degradation was observed when CTDSP1 function was inhibited. Immunoblot analysis of topoI and reduction in the florescence intensity of topoI-EGFP clearly demonstrated enhanced topoI degradation in CTDSP1-deficient cells. Others and we have (change this) shown that enhanced rates of topoI degradation cause irinotecan resistance. To further validate this pathway, we asked if CTDSP1-mediated activation of DNA-PKcs and rapid degradation of topoI causes irinotecan resistance. Cells that were treated with rabeprazole did show irinotecan resistance in both cell survival and clonogenic assay.

CTDSP1 is a member of phosphatases that rely on the DxDx motif and $Mg^{2+}$ to catalyze the phosphoryl-transfer [20, 29]. Rabeprazole binds to a unique hydrophobic pocket of CTDSP1, located in the insertion domain and adjacent to the DxDx motif. Digestive symptoms are the most feared complications of chemotherapy [30, 31] and a proton pump inhibitor (PPI) like rabeprazole, is frequently prescribed to patients. PPI. including rabeprazole, was recently reported to enhance the antitumor effects of both docetaxel-cisplatin combinations [32] and 5-FU [33] in cells. PPI can restore drug sensitivity in drug-resistant cells by preventing the increase of extracellular and lysosomal pH and increasing the cytoplasmic retention of doxorubicin [34]. However, another study highlighted a potential adverse effect of PPI on overall survival in colorectal cancer [35]. Treatment with a PPI may result in hypochlorhydria in the stomach, which causes hypersecretion of the hormone gastrin from the gastric antrum [36]. Moreover, *in vitro* studies have shown that hypergastrinemia promotes cell proliferation and migration, and inhibits apoptosis [37–39]. The relationship between PPI therapy and chemosensitivity is not clear. Notably, none of these studies examined the impact of rabeprazole on irinotecan sensitivity. This study is the first to demonstrate that rabeprazole inhibited CTDSP1 activity and caused resistance to irinotecan. Based on our findings, the concurrent use of rabeprazole should be avoided in cancer patients under treatment with irinotecan. If PPI therapy cannot be discontinued due to the severity of the gastrointestinal symptoms, alternative PPIs should be employed.

This study has some limitations. All analyzed patients underwent irinotecan chemotherapy as a second-line regimen and this analysis was retrospective. Prospective studies focusing on 1st-line irinotecan regimens would be necessary to definitively establish the relationship between rabeprazole and irinotecan.

In conclusion, we showed that CTDSP1 and its inhibitor, rabeprazole, play important roles in topoI regulation, particularly in response to irinotecan. This study clearly indicated that rabeprazole induces irinotecan resistance by enhancing proteasomal degradation of topoI and may not be a suitable PPI for cancer patients receiving irinotecan-based therapy.

## Supporting information

**S1 Fig. The pictures that performed clonogenic assays of HCT116 and HT29 cells.** HCT116 and HT29 cells were treated with various concentrations of SN-38 and clonogenic assays were performed to determine the relative number of colonies.
(TIF)

**S1 Table. Sequences of plasmids used for genome editing.**
(DOCX)

**S1 Data.**
(ZIP)

## Acknowledgments

The authors thank S. Tsurumaru and A. Nakamura for technical assistance with the experiments.

## Author Contributions

**Conceptualization:** Hiroya Matsuoka, Eiji Oki, Ajit K. Bharti.

**Data curation:** Hiroya Matsuoka, Ajit K. Bharti.

**Formal analysis:** Hiroya Matsuoka, Emma J. Swayze, Elizabeth C. Unan, Joseph Mathew, Quingjiang Hu, Yasuo Tsuda, Yuichiro Nakashima, Hiroshi Saeki, Ajit K. Bharti.

**Investigation:** Hiroya Matsuoka.

**Methodology:** Hiroya Matsuoka, Koji Ando, Quingjiang Hu, Yasuo Tsuda, Yuichiro Nakashima, Hiroshi Saeki, Eiji Oki, Ajit K. Bharti.

**Project administration:** Hiroya Matsuoka, Koji Ando.

**Supervision:** Hiroshi Saeki, Eiji Oki, Ajit K. Bharti, Masaki Mori.

**Validation:** Hiroya Matsuoka, Koji Ando, Ajit K. Bharti.

**Visualization:** Hiroya Matsuoka.

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
