## [Decision Letter · Decision Letter 0]

17 Feb 2020

PONE-D-20-00259

CTDSP1 inhibitor rabeprazole regulates DNA-PKcs dependent topoisomerase I degradation and irinotecan drug resistance in colorectal cancer

PLOS ONE

Dear Dr. Koji Ando,

Thank you for submitting your manuscript to PLOS ONE. After careful consideration, we feel that it has merit but does not fully meet PLOS ONE’s publication criteria as it currently stands. Therefore, we invite you to submit a revised version of the manuscript that addresses the points raised during the review process.

This manuscript identified CTDSP1 as a primary upstream regulator of DNA-PKcs in response to topo I inhibitors. The article is interesting. However, there are some major concerns for this study. The description for M & M is unclear. It have to be wrote the detail design. In addition, The quality for the results (Western blot) is poor. It have to be improved and followed the instruction for PLOS ONE to uploaded the original image during the revising submission. Moreover, the statistic is lacked in the manuscript. Overall, the article have to improve to fit the scientific requirement.

We would appreciate receiving your revised manuscript by Apr 02 2020 11:59PM. To enhance the reproducibility of your results, we recommend that if applicable you deposit your laboratory protocols in protocols.io, where a protocol can be assigned its own identifier (DOI) such that it can be cited independently in the future. For instructions see: http://journals.plos.org/plosone/s/submission-guidelines#loc-laboratory-protocols

We look forward to receiving your revised manuscript.

Kind regards,

Yu-Jia Chang

Academic Editor

PLOS ONE

Journal Requirements:

Reviewers' comments:

Reviewer's Responses to Questions

**Comments to the Author**

1. Is the manuscript technically sound, and do the data support the conclusions?

Reviewer #1: Partly

Reviewer #2: Partly

Reviewer #3: Partly

2. Has the statistical analysis been performed appropriately and rigorously? 

Reviewer #1: No

Reviewer #2: N/A

Reviewer #3: No

3. Have the authors made all data underlying the findings in their manuscript fully available?

Reviewer #1: Yes

Reviewer #2: Yes

Reviewer #3: No

4. Is the manuscript presented in an intelligible fashion and written in standard English?

Reviewer #1: No

Reviewer #2: Yes

Reviewer #3: No

5. Review Comments to the Author

Reviewer #1: This study claimed that CTDSP1 inhibitor rabeprazole regulates DNA-PKcs dependent topoisomerase I degradation and irinotecan drug resistance in colorectal cancer. The findings are likely useful for overcoming irinotecan resistance in colorectal cancer, as well as other solid tumors. However, experimental designs and descriptions are need to be improved. I have some comments as follows.

Major comments:

1. In Figure 1, the authors should quantify the intensity of each blots compared to beta-actin protein levels. In Figure 1B, the authors should determine the phosphorylated Topo I levels.

2. The authors treated cells with 2.5 �M SN-38 in Figure 1B but treated cells with SN-38 at concentrations lower than 7.5 nM in Figure 1C. Similar views also found in Figures 2, 3 and 5. Is it a typo? Otherwise, it is hard to believe that Topo I protein levels determine the therapeutic effectiveness of SN-38. Unless the authors show that Topo I protein degradation is visible at the treatment with SN-38 at low nM range.

3. In Figure 2B, the remarkable protein degradation should be detected in CTDSP1 knockdown cells at time 0 in comparison with control cells. However, it seems to be not. Please describe it.

4. The experimental procedure to generate Topo I-EGFP fusion by CRISPR/Cas9 system is not clearly presented. Please provide the sequence of sgRNA used in this study and sequencing result for the gene fusion.

5. What is the protein band for pDNA-PKcs in Figure 4B?

6. In Figure 5A, the protein levels of Topo 1 at time 0 seem to be not equal. Why?

7. In Table 1, if the clinical data from a small sample size is capable of supporting the conclusion that rabeprazole had an effect on the response to irinotecan in clinical settings?

Minor comments:

1. The authors should describe statistical tools they used in MM section

2. I found many typos and grammar errors in the manuscript. So, English editing of this manuscript should be needed.

Reviewer #2: This manuscript identified CTDSP1 as a primary upstream regulator of DNA-PKcs in response to topo I inhibitors. The authors used HCR116 and HT29 to identify the effect of CTDSP1 in irinotecan response. The results imply that CTDSP1 is related to drug sensitivity of irinotecan and rabeprazole treatment, resulting in drug resistant occurrence. In general, the idea of this manuscript is interesting. In order to improve the quality of this study, several suggestions are given in the following list.

1: The expressions of CTDSP1 and Topo I should be evaluated and quantified in western blot and Q-PCR analysis. This includes figure 1, 2, 3 and 5.

2: The appropriate internal control of pDNAPKcs is total form of DNAPKcs. Please add this important data with individual pDNAPKcs western blot.

3: p-ATM, ATM, p-ATR, ATR, gimma H2AX and XRCC4 protein expressions are always changed, followed by the activation of pDNAPKcs. The authors should also evaluate these proteins.

4: Several expressions of control are not equally presented. I would suggest to repeat these data. Example: figure 2b, time point 0 of control and siCTDSP1. Figure 2b, time point 0 of control and CTDSP1. figure 5, time point 0 of 0, 10 and 20uM drug treatment.

5: Immunofluorescence (IF) is not the appropriate method to quantify protein expression changes. All the protein expression of IF should be measured by flow cytometry, in order to understand both protein expression and cell population.

Reviewer #3: The presented study investigated the role of CTDSP1 in topoisomerase I associated irinotecan resistance in human colorectal cancer cells and effects of rabeprazeole, a CTDSP1 inhibitor, for reducing irinotecan resistance. Although experiments performed in this study were well organized and conducted, this manuscript was not prepared in an appropriate academic format with a plenty of missing information. This manuscript therefore is required a major revision before the acceptance is considered.

Comments:

1. Instruments, chemicals and reagents used in experiments shall be clearly described with information manufacturers including name of manufacturers and location in the section of materials and methods. Some of them only stated in results and figure legends such as Leica SP5 confocal microscope and EGFP.

2. Line 182, cell density for each cell type shall be described clear.

3. Replication numbers of each experiment shall be described by using proper statistical analysis to verify the differences among treatment conditions. Particularly the results of immunoblottings shall contain quantification data (E.g. Measurement of intensity) to compare the differences between cell types, treatments and experimental conditions with statistical analysis as well.

4. Methodology of statistics for this manuscript is completely missed. Statistical analysis for both in vitro and in vivo studies shall be clearly described in an independent paragraph of materials and methods. Authors claimed many significant differences and effects among experiments but gave no statistical results for verification.

5. This manuscript was not written in a proper academic English and organization. For example, the aims of study was not clearly described in introduction, and too many active voices and questing among writing. Professional English editing service therefore is required to make this manuscript more readable.

6. PLOS authors have the option to publish the peer review history of their article (what does this mean?). If published, this will include your full peer review and any attached files.

Reviewer #1: No

Reviewer #2: Yes: Chia-Hwa Lee

Reviewer #3: No

---

## [Author Response · Author response to Decision Letter 0]

5 Apr 2020

We also appreciate the time and effort you and each of the reviewers have dedicated to providing insightful feedback on ways to strengthen our paper. Thus, it is with great pleasure that we resubmit our article for further consideration. We have incorporated changes that reflect the detailed suggestions you have graciously provided.

---

## [Decision Letter · Decision Letter 1]

11 May 2020

PONE-D-20-00259R1

CTDSP1 inhibitor rabeprazole regulates DNA-PKcs dependent topoisomerase I degradation and irinotecan drug resistance in colorectal cancer

PLOS ONE

Dear Dr. Ando_Koji,

Thank you for submitting your manuscript to PLOS ONE. After careful consideration, we feel that it has merit but does not fully meet PLOS ONE’s publication criteria as it currently stands. Therefore, we invite you to submit a revised version of the manuscript that addresses the points raised during the review process.

ACADEMIC EDITOR: There are some major concerns which need to be improved. Please check the reviewers' question. 

We would appreciate receiving your revised manuscript by Jun 25 2020 11:59PM. To enhance the reproducibility of your results, we recommend that if applicable you deposit your laboratory protocols in protocols.io, where a protocol can be assigned its own identifier (DOI) such that it can be cited independently in the future. For instructions see: http://journals.plos.org/plosone/s/submission-guidelines#loc-laboratory-protocols

We look forward to receiving your revised manuscript.

Kind regards,

Yu-Jia Chang

Academic Editor

PLOS ONE

Reviewers' comments:

Reviewer's Responses to Questions

**Comments to the Author**

1. If the authors have adequately addressed your comments raised in a previous round of review and you feel that this manuscript is now acceptable for publication, you may indicate that here to bypass the “Comments to the Author” section, enter your conflict of interest statement in the “Confidential to Editor” section, and submit your "Accept" recommendation.

Reviewer #1: (No Response)

Reviewer #2: (No Response)

Reviewer #3: All comments have been addressed

2. Is the manuscript technically sound, and do the data support the conclusions?

Reviewer #1: Yes

Reviewer #2: Partly

Reviewer #3: Yes

3. Has the statistical analysis been performed appropriately and rigorously? 

Reviewer #1: Yes

Reviewer #2: Yes

Reviewer #3: Yes

4. Have the authors made all data underlying the findings in their manuscript fully available?

Reviewer #1: Yes

Reviewer #2: No

Reviewer #3: Yes

5. Is the manuscript presented in an intelligible fashion and written in standard English?

Reviewer #1: Yes

Reviewer #2: No

Reviewer #3: Yes

6. Review Comments to the Author

Reviewer #1: The authors have fully answered my questions. I suggest the manuscript is acceptable for publication.

Reviewer #2: This is the second-round review of this manuscript. Unfortunately the authors failed to appropriately response the comment from the first-round review.

4. The experimental procedure to generate TopoI-EGFP fusion by CRISPR/Cas9 system is not clearly

presented. Please provide the sequence of sgRNA used in this study and sequencing result for the gene

fusion.

A well hTOP1 gene edit efficiency analysis and protein expression analysis should be performed in this manuscript. A previous study may be taken as an example. HDAC1,2 Knock-Out and HDACi Induced Cell Apoptosis in Imatinib-Resistant K562 Cells. Int J Mol Sci. 2019 May 8;20(9). pii: E2271. doi: 10.3390/ijms20092271.

With the second review, some critical issues raised in the revised manuscript and the comments need to be appropriately responded.

1: The photos of clonogenic assay in figure 1C, 3C should be attached in supplementary data.

2: It is very confusing of the following description “A comparison of two CRC cell lines, HCT116 and HT29, indicated higher expression of CTDSP1 in HCT116. A comparison of two CRC cell lines, HCT116 and HT29, indicated higher expression of 3 CTDSP1 in HCT116. We also observed minimal proteasomal degradation and HCT116 and HCT116 cells are significantly more sensitive to SN-38.” Page 9, line 2-4.

3: The significant TopoI protein expression drop is occurred after 2.5 μM SN-38 treatment in HCT116 cell (figure 1B). However, the bar figure doesn’t reflect this protein change. The results from blot and bar figure are not consistent.

4: The protein expression blot of topoI-GFP protein in figure 2C and 5B should be revealed in figures.

5: The CTDSP1 expression in figure 1B, 2B, 3B, 4B, and 5A should also be revealed.

6: In figure 3B, a very significant topoI increase was observed in cells overexpressing CTDSP1, the description in result section is misleading.

7: A western blot result of pDNAPKcs expression in figure 4C should be revealed in the same figure.

8: The orders of the cell input in blot should consistent in this manuscript.

For example: figure 2A is different from other figures (control, si or overexpression).

9: IRB approval number is nor provided in the revised manuscript.

Reviewer #3: This reviewer is satisfied with this revised manuscript and suggest an acceptance decision to be considered by PLOS One.

7. PLOS authors have the option to publish the peer review history of their article (what does this mean?). If published, this will include your full peer review and any attached files.

Reviewer #1: No

Reviewer #2: Yes: Chia-Hwa Lee

Reviewer #3: No

---

## [Author Response · Author response to Decision Letter 1]

31 May 2020

Thank you for inviting us to submit a revised draft of our manuscript entitled, “CTDSP1 inhibitor rabeprazole regulates DNA-PKcs dependent topoisomerase I degradation and irinotecan drug resistance in colorectal cancer.” to PLOS ONE. We also appreciate the time and effort you and each of the reviewers have dedicated to providing insightful feedback on ways to strengthen our paper. Thus, it is with great pleasure that we resubmit our article for further consideration. We have incorporated changes that reflect the detailed suggestions you have graciously provided.

---

## [Editor Report · Decision Letter 2]

1 Jul 2020

CTDSP1 inhibitor rabeprazole regulates DNA-PKcs dependent topoisomerase I degradation and irinotecan drug resistance in colorectal cancer

PONE-D-20-00259R2

Dear Dr. Koji Ando

We’re pleased to inform you that your manuscript has been judged scientifically suitable for publication and will be formally accepted for publication once it meets all outstanding technical requirements.

Kind regards,

Yu-Jia Chang

Academic Editor

PLOS ONE
---

## [Editor Report · Acceptance letter]

9 Jul 2020

PONE-D-20-00259R2 

CTDSP1 inhibitor rabeprazole regulates DNA-PKcs dependent topoisomerase I degradation and irinotecan drug resistance in colorectal cancer 

Dear Dr. Ando:

I'm pleased to inform you that your manuscript has been deemed suitable for publication in PLOS ONE. Congratulations! Your manuscript is now with our production department. 

Kind regards, 

on behalf of

Dr Yu-Jia Chang 

Academic Editor

PLOS ONE